# COVID-19: Diabetes Perspective—Pathophysiology and Management

**DOI:** 10.3390/pathogens12020184

**Published:** 2023-01-25

**Authors:** Siva Dallavalasa, SubbaRao V. Tulimilli, Janhavi Prakash, Ramya Ramachandra, SubbaRao V. Madhunapantula, Ravindra P. Veeranna

**Affiliations:** 1Center of Excellence in Molecular Biology and Regenerative Medicine (CEMR) Laboratory (DST-FIST Supported Centre), Department of Biochemistry (DST-FIST Supported Department), JSS Medical College, JSS Academy of Higher Education and Research (JSS AHER), Mysuru 570015, India; 2Department of Biochemistry, Council of Scientific and Industrial Research (CSIR)-Central Food Technological Research Institute (CFTRI), Mysuru 570020, India; 3Leader, Special Interest Group in Cancer Biology and Cancer Stem Cells (SIG-CBCSC), JSS Medical College, JSS Academy of Higher Education and Research (JSS AHER), Mysuru 570015, India

**Keywords:** COVID-19, SARS-CoV-2, diabetes, glucocorticoids, cytokines, ARDS

## Abstract

Recent evidence relating to the impact of COVID-19 on people with diabetes is limited but continues to emerge. COVID-19 pneumonia is a newly identified illness spreading rapidly throughout the world and causes many disabilities and fatal deaths. Over the ensuing 2 years, the indirect effects of the pandemic on healthcare delivery have become prominent, along with the lingering effects of the virus on those directly infected. Diabetes is a commonly identified risk factor that contributes not only to the severity and mortality of COVID-19 patients, but also to the associated complications, including acute respiratory distress syndrome (ARDS) and multi-organ failure. Diabetic patients are highly affected due to increased viral entry into the cells and decreased immunity. Several hypotheses to explain the increased incidence and severity of COVID-19 infection in people with diabetes have been proposed and explained in detail recently. On the other hand, 20–50% of COVID-19 patients reported new-onset hyperglycemia without diabetes and new-onset diabetes, suggesting the two-way interactions between COVID-19 and diabetes. A systematic review is required to confirm diabetes as a complication in those patients diagnosed with COVID-19. Diabetes and diabetes-related complications in COVID-19 patients are primarily due to the acute illness caused during the SARS-CoV-2 infection followed by the release of glucocorticoids, catecholamines, and pro-inflammatory cytokines, which have been shown to drive hyperglycemia positively. This review provides brief insights into the potential mechanisms linking COVID-19 and diabetes, and presents clinical management recommendations for better handling of the disease.

## 1. Introduction

Coronavirus disease-19 (COVID-19) is an epidemic and caused by Severe Acute Respiratory Syndrome-Coronavirus (SARS-CoV)-2. SARS-CoV-2 is a single-stranded, positive sense, enveloped RNA virus, which infects both humans and animals [1]. Earlier, coronaviruses caused two major outbreaks globally; whereas the first outbreak in 2002–2003 was due to SARS-CoV with a 10% mortality rate, the second outbreak in 2012 was due to the severe spread of Middle East Respiratory Syndrome Corona Virus (MERS-CoV) with a 36% mortality [2]. Coronaviruses are composed of four structural proteins, namely, spike (S), membrane (M), nucleocapsid (N), and envelope (E) proteins. The S protein plays a major role in the virus’s attachment to the host cell receptor on the cell membrane and entry into the host cells through its S1 and S2 subunits [3]. The host cell surface receptor for SARS-CoV and SARS-CoV-2 is angiotensin-converting enzyme 2 (ACE-2), while the receptor for MERS-CoV is dipeptidyl peptidase 4 (DPP4) [4]. ACE-2 receptors are expressed predominantly in the upper respiratory tract (both type I and II alveolar epithelial cells), endothelial cells, kidney tubular epithelium, heart, enterocytes, and pancreas [5]. The presence of cathepsin-L proteases and alkalinity in the endosome favors the release of the viral genome into the cytosol [6]. The virus replicates in target cells and forms mature virions. Cells containing mature virions undergo apoptosis and activate pro-inflammatory cytokines [7]. Further, T Helper (Th1) regulates the antigen activity and immunity against pathogens by producing interferon-gamma (IFNγ). Th17 cells prompt macrophages and mobilize neutrophils by producing interleukins (IL) such as interleukin-17 (IL-17), IL-21, and IL-22. Subsequently, lymphocytes such as CD3, CD4, and CD8 T cells undergo apoptosis due to the infection of immune cells and cause lymphocytopenia [8]. Moreover, secretion of unusually high inflammatory cytokines such as IL-6 and tumor necrosis factor (TNF), chemokine CXC-chemokine ligand 10 (CXCL10), and CC-chemokine ligand 2 (CCL2) leads to “cytokine storm”, which causes the inhibition of innate immunity [9]. Further, the cytokine storm leads to hyper-inflammation, which triggers the induction of endothelial HA-synthase-2 (HAS2) in alveolar epithelial cells (type 2), and fibroblasts and accelerates the synthesis of hyaluronan (HA) [10]. The synthesized HA binds to water up to 1000 times of its molecular weight, which causes the accumulation of fluid surrounding alveolar epithelial cells. Consequently, the exchange of oxygen and carbon dioxide is affected, leading to hypoxia in the lung and in other organs contributing to multi-organ failure and death [11].

Diabetes, a major metabolic disorder, is one of the leading causes of mortality and morbidity throughout the world [12]. Several studies have shown that diabetics infected with Influenza A (H1N1), MERS-CoV, and SARS-CoV exhibit severe disease symptoms and succumb to death compared to non-diabetic individuals [13,14,15]. Similarly, infection with SARS-CoV-2 is also associated with severe complications in diabetics. For example, SARS-CoV-2 infection led to the release of unusually high glucocorticoids and catecholamines, which further elevated blood glucose [16]. The hyperglycemic condition, in turn, recruits the pro-inflammatory monocytes, enhances platelet reactivity, and therefore contributes to increases in the number of cardiovascular deaths in diabetic individuals [17].

The practices used for treating the SARS-CoV-2 infected individuals with existing diabetes are solely different from those of the general population [18]. For example, some of the anti-diabetic drugs such as biguanides, SGLT2 inhibitors, DPP-4 inhibitors, sulfonylureas, and insulin, which have been commonly administered for the management of type-2 diabetes, are reported to exacerbate the clinical conditions in COVID-19. However, additional studies are warranted as the existing evidence is contradictory. In a recent study, DPP-4 inhibitors are suggested for hospitalized COVID-19 patients suffering from diabetes [19], but the use of sulfonylureas and metformin against SARS-CoV-2-infected diabetic patients is still controversial [20]. Even though earlier studies have demonstrated the use of anti-diabetics drugs for COVID-19 patients due to their minimal risk even after long exposure time, relatively low cost, and easy availability, further studies are warranted to confidently recommend these drugs for treating diabetic individuals infected with SARS-CoV-2 [21]. Therefore, in this article, we have reviewed various research articles demonstrating the therapeutic effects of the most commonly used anti-diabetic drugs in individuals infected with SARS-CoV-2.

## 2. Enhanced Fatality in Diabetic Subjects Infected with SARS-CoV-2

Several clinical studies have reported that diabetes is one of the major causes of co-morbidities associated with the SARS-CoV-2 infection [22,23,24]. A study by Kumar et al. (2020) showed a twofold increase in mortality and severity due to SARS-CoV-2 infection in diabetic patients compared to non-diabetics [25]. A study conducted by the Chinese Centre for Disease Control and Prevention finds that COVID-19 mortality was 7.3% in diabetic patients compared to 2.3% in non-diabetic patients [26]. Similarly, a study from Hong Kong showed that among COVID-19-positive cases, 42.3% were diabetics [22]. In another study, Xiaobo Yang and colleagues (2020) reported a predominance of diabetic patients among ICU admissions, i.e., 32 out of 52 ICU patients had diabetes [27]. A study by Guan et al. (2020) also showed that out of 1099 patients with COVID-19, 173 had severity with co-morbidities, of which 16.2% had diabetes, 23.7% reported hypertension, 5.8% had coronary heart diseases, and 2.3% reported cerebrovascular disease [28]. A recent study stated that, out of 140 patients with COVID-19, 12% had diabetes, and 30% had hypertension [29]. One of the multivariate analyses found that diabetes has an independent relationship with increased mortality rate due to COVID-19 [30]. Potential reasons for enhanced severity and mortality due to COVID-19 in diabetics include (a) irregular/missed blood glucose (BG) monitoring; (b) corticosteroid therapy; (c) inappropriate/discontinuation of angiotensin II receptor blockers (ARBs) and angiotensin-converting enzyme inhibitors (ACEIs); and (d) isolation from physicians.

## 3. COVID-19 and Diabetes

Recent multi-centric studies have shown that the severity of COVID-19 is much higher in diabetes patients [31,32,33]. This is in part due to (a) insulin resistance; (b) hyperglycemia, which promotes the synthesis of pro-inflammatory cytokines and advanced glycation end products (AGEs); (c) oxidative stress; (d) production of adhesion molecules that mediate inflammation of the tissue; and (e) a robust pro-inflammatory response [31,32,33]. Abnormally delayed-type hypersensitivity reactions complement activation dysfunction, inhibit lymphocyte proliferative response to various stimuli, and impair macrophage and neutrophil functions in patients with diabetes [34,35]. The pulmonary epithelial cells, which were exposed to hyperglycemia, had an augmented viral infection as well as replication, indicating the role of hyperglycemia in the in vivo enhancement of viral infections [36,37]. SARS-CoV-2 binds to ACE-2 receptors expressed in the cells of the pancreas, adipose tissue, and intestine [38]. Therefore, it is possible that SARS-CoV-2 infection might cause alterations in glucose metabolism and contribute to the augmented severity of COVID-19.

### 3.1. COVID-19 and Renin–Angiotensin System on Diabetes

The renin–angiotensin system (RAS) blockers may enhance COVID-19 by boosting the expression of ACE2 [39]. Since SARS-CoV-2 utilizes ACE2 to penetrate host cells, the RAS-blockers can further trigger the spread of the virus in diabetics [40]. ACE-2 is majorly expressed in the upper respiratory tract, in the lungs, and in both type I and II alveolar epithelial cells, endothelial cells, kidney tubular epithelium, heart, enterocytes, cerebral neurons, intestines, immune cells, pancreas, and endothelium of veins and arteries [5]. Further, replication of SARS-CoV-2 leads to the spread of mature virions, which upon exposure to the host immune system leads to the production of interleukins, induces apoptosis of lymphocytes, and inhibits innate immunity. The resulting cytokine storm leads to hyperinflammation and multi-organ failure [41]. In certain studies, it is shown that an RAS blockade could reduce the risk of COVID-19 severity in hypertensive patients [42]. Another study pointed out that the interaction of SARS-CoV-2 with ACE2 causes dysregulation in the host response to viral infection. This is because ACE2 regulates the production of angiotensin II (Ang II), and SARS-CoV-2, by destroying ACE2, allowing the unrestricted development of Ang II. However, the role of Ang II in tissue response to injury is mostly governed by its metabolism and conversion to Ang-(1–7) by the enzyme ACE2 [43]. Ang-(1–7) exhibits effects that are opposite to those mediated by Ang II-AT1R. As a result of the SARS-CoV-2-infection-induced degradation of ACE2, there is a lack of control over Ang II’s pro-inflammatory activities and tissue damage. As a result, the pathogenesis of COVID-19 is caused by the host’s response to SARS-CoV-2 [43]. The clinical processes that are intensified after SARS-CoV-2 infection in people with diabetes mellitus are shown in Figure 1.

### 3.2. COVID-19 and Insulin Resistance

Insulin resistance has serious health consequences that affect vasculature, the heart, the brain, and the kidneys [44]. COVID-19 causes insulin resistance in patients, resulting in chronic metabolic abnormalities that were not present before in these patients [45]. Insulin is a hormone secreted from islets of Langerhans in the pancreas. Insulin activates the transportation of glucose to muscle and adipose tissues [46]. Insulin resistance is due to decreased tissue sensitivity to insulin and the inability of the pancreas to secrete sufficient insulin for blood glucose regulation for physiological functions [47,48]. ACE-2 converts angiotensin-2, (which is a vasoconstrictor, pro-fibrotic, and pro-inflammatory molecule) into angiotensin 1–7, to cause vasodilation [49]. SARS-CoV-2 infection decreases ACE2 expression and causes an increase in Ang II activity and insulin resistance [45]. In addition, an increased immunological response was also reported due to SARS-CoV-2 infection [45]. Recent reports stated that pancreatic β-cells are damaged in patients with SARS-CoV-2 infection, and individuals who had not received glucocorticoids were reported to have significantly increased fasting glucose levels compared to non-SARS pneumonia patients [50]. The viral infection leads to pancreatic β-cells destruction and disturbs the homeostasis of blood glucose levels in patients with T2DM [51]. These alterations in glucose metabolism and pancreatic infections with SARS-CoV-2 might result in the onset of T2DM or type 1 diabetes [52]. A recent study has shown that individuals infected with SARS-CoV-2 had about a 40% increase in diabetes, but the reasons for this increase are not fully known [53].

### 3.3. COVID-19 Mediated Inflammatory Responses Pertaining to Diabetes

Cytokine storm is one of the major disturbances reported in COVID-19 patients. Kulcsar et al. (2019) have reported that diabetic male mice showed alterations in CD4 T helper cell counts and elevated IL17α [54]. In MERS experimental model, mice with diabetes showed lesser inflammatory macrophages, CD4 T cells, and very low levels of CCl2 and CXCl10 when compared to control mice [54]. In patients with COVID-19, not only were the CD4 and CD8 cells low, but they had increased pro-inflammatory cytokines [55]. In patients with DM, an abnormal antiviral interferon response and delayed activation of Th1/Th17 were also reported along with increased inflammatory reactions [41]. Mild COVID-19 induces a pro-inflammatory response with higher amounts of inflammatory cytokines, and inducible protein-10 (IP10) expression, which leads to lesser insulin sensitivity. DM suppresses neutrophil chemotaxis, phagocytosis, and intracellular killing of microbes; hence, in patients with diabetes, an initial delay was observed in the activation of Th1 cell-mediated immunity, abnormalities in adaptive immunity, as well as a late hyperinflammatory response [56,57,58].

### 3.4. Role of Glycemic Control in Determining Infection Rates

Type-2 diabetes patients are characterized by weak immune responses due to poor glycemic control [59]. The immune responses of Type-2 diabetics to SARS-CoV-2 infection are controversial [60]. Raffaele Marfela et al. (2022) showed poor glycemic effects in COVID-19-infected vaccinated people, and reported lower immune responses and decreased HbA1c levels in T2D patients, who are vaccinated with m-RNA-BNT162B2 [61]. Bergwerk, M. et al. stated that lower titers of neutralizing antibodies play a crucial role in the infection period [62]. Based on recent study reports, several factors, which include sex, obesity, cardiovascular disease, and type-2 diabetes, are frequently occurring conditions in fully vaccinated patients and are responsible for devolving the COVID-19 infection [63,64]. Type-2 diabetic patients are mainly characterized by decreased immune responses to naturally occurring infections and vaccination [65,66]. Previous studies stated that COVID-19 vaccination showed weak immunity in type-2 diabetic patients with poor glycemic control when compared with normal diabetic patients [60]. Therefore, effective glycemic control helps in launching strong immune responses against SARS-CoV-2 virus, thereby reducing the complications of COVID-19.

### 3.5. COVID-19 and Other Complications Concerning Diabetes

In addition to insulin resistance, inflammatory responses, the renin–angiotensin system, and other factors such as clotting factors, hemoglobin, obesity, hypokalemia, and hypovitaminosis D were also affected by COVID-19 in diabetic patients. In T2DM, coagulation and fibrinolysis imbalance are often observed [67,68]. The amount of clotting factors increase, while the fibrinolytic system is inhibited. Hyperglycemia and insulin resistance are linked with enhanced activation and aggregation of platelets [69]. These conditions result in a hypercoagulable pro-thrombotic state [17]. SARS-CoV-2 infection impairs the ability of hemoglobin to carry oxygen as the b1-chain of hemoglobin is attacked by the non-structural proteins of SARS-CoV-2, which causes the separation of iron from the porphyrin ring and, hence, impairs the oxygen-carrying capacity of hemoglobin. Reports have also suggested that SARS-CoV-2 has more affinity towards glycated hemoglobin than the non-glycated form [27]. Obesity is associated with T2DM and adiposities in the case of SARS-CoV-2.

Very high viral titers might enhance the hyper-immune reaction, thereby aggravating insulin resistance [70]. COVID-19 is also associated with hypokalemia, which downregulates ACE-2 receptors on lung epithelial cells leading to an increase in the secretion of aldosterone [71]. Hypokalemia can also worsen glucose levels in patients suffering from T1DM and T2DM [72]. Hypovitaminosis D might also cause insulin resistance, and adequate vitamin D supplementation is demonstrated to improve insulin receptivity [73]. Hence, vitamin D deficiency can worsen insulin resistance in patients infected with SARS-CoV-2 [74]. Rao et al. (2020) observed a link between DM and the upregulation of furin (cellular protease), which is involved in virus entry [75]. Hence, the rate of infection is more in patients with DM. Moreover, T2DM is correlated with a worse outcome for COVID-19. A study by Zhu et al. (2020) showed that among 7300 cases of COVID-19, T2DM is associated with an increase in the mortality rate, but diabetic people with well-monitored blood glucose have significantly recovered [76].

## 4. Potential Mechanisms Increasing the Susceptibility for COVID-19 in Patients with DM

Diabetes is one of the main risk factors for many infections that include H1N1, influenza, and MERS-CoV [54,77,78]. For instance, out of many co-morbidities such as older age, genetic history, being male, and many underlying pathological conditions, such as mainly chronic lung disease, hypertension, and cardiovascular disease, diabetes is associated with greater severity of the infections. Diabetes is mainly associated with mortality and severity among COVID-19 patients [79,80]. Kumar (2020) reported that diabetic patients with COVID-19 are associated with a twofold increase in mortality and severity of COVID-19 than non-diabetic individuals [81].

Several factors are responsible for the increased severity of infections with SARS-CoV-2 in DM. Recent studies have stated that high affinity for cellular binding and entry of virus; higher viral titer; relatively low functioning T cells, hence, lower viral clearance; susceptibility to hyper inflammation and cytokine storm syndrome; co-morbidities associated with T2DM such as CVD, NAFLD, hypertension, and obesity exposes the diabetics more vulnerable to virus infections [15,82,83].

Other risk factors may include increased expression of ACE-2, upregulation of furin, impaired T-cell function, and increased levels of cytokines [54,75,84,85]. The mechanisms that are increasing the susceptibility of diabetics to COVID-19 are represented in Figure 2.

### 4.1. COVID-19 Viral Load

Viral load is an important measure, which indicates the number of viral particles in a given volume of body fluid or a cell. It is a key parameter required to assess the infectivity and severity of the disease. Since the receptor for SARS-CoV-2 is expressed in tissues of the heart, lung, kidney tubules, blood vessels, and the luminal surface of the small intestine, there is a possibility for this virus to attack these cells and increase the viral load enormously [86]. Increased expression of ACE2 was reported in several tissues of the lung, kidney, and heart in diabetic mice compared to healthy non-diabetic ones [87]. Higher expression of ACE2 in diabetic patients strongly facilitates the entry of SARS-CoV-2 into cells [88,89]. Furthermore, commonly prescribed pharmacological agents to diabetics that include GLP-1 agonists, ACE inhibitors, statins, and anti-hypertension agents are generally known to up-regulate the expression of ACE2 in various cell types [90]. Clinically, diabetic individuals with improved glycemic control are reported to be less susceptible to COVID-19 illness [91]. Thus, the severity of COVID-19 is higher in uncontrolled diabetic patients compared to the ones with controlled diabetes.

### 4.2. SARS-CoV-2 Dysregulates Immune Responses and Causes Cytokine Storm

Innate and adaptive immune systems play a crucial role in clearing the viral load. One of the common complications found in diabetic individuals is dysregulation of the immune system, which is associated with an increase in an inflammatory response [92]. Potential reasons for the enhanced severity of SARS-CoV-2 in diabetics are (a) dysregulation of immune responses, (b) abnormal cytokine responses, and (c) irregular immune cell numbers [92]. Moreover, it has been reported that the elevation of glucose levels may suppress the antiviral responses [92,93,94]. Further, the severity of COVID-19 could also be due to delayed interferon-gamma response and is associated with a prolonged hyperinflammatory state as well as lower CD4+ and CD8+ cell numbers [95]. Individuals with diabetes have shown variations in the components of the innate immune system and exhibit impaired chemotaxis and phagocytosis [96,97]. Individuals with diabetes have reduced NK cell activity than individuals with normal glucose tolerance or prediabetes [98]. Similarly, Kratz (2014) reported the presence of more pro-inflammatory M1 macrophages in individuals with T2D [99]. A low-grade inflammation was also reported in patients with diabetes due to the imbalance between Th1/Th2 [100]. The reason behind the cytokine storm that increased the severity of COVID-19 is (a) hyperinflammatory response and (b) delayed interferon-gamma response in individuals with diabetes.

### 4.3. Alveolar Dysfunction

Various structural changes, such as increased vasculature permeability and decreased gas exchange, have been reported in the lungs of patients suffering from diabetes [101]. Individuals with diabetes have shown pulmonary complications that led to an increase in mechanical ventilation during COVID-19 [102]. One of the studies conducted in New York City with 5700 patients reported that the patients who were dead with diabetes had received invasive mechanical ventilation during their treatment compared to the ones without diabetes [103]. Similarly, one of the findings showed that T2DM is associated with an increase in the occurrence of ventilator-associated pneumonia in mechanically ventilated adult trauma patients [104]. Individuals with diabetes show significantly thicker endothelial capillary basal lamina and alveolar epithelia than individuals without diabetes [105]. Hence, individuals with diabetes exhibit impaired respiratory function in association with the susceptibility of SARS-CoV-2 to infect lung tissue cells, which may provoke pulmonary complications of COVID-19.

### 4.4. Endothelial Dysfunction

Virus-induced lytic programmed cell death is associated with inflammation and vascular damage, which are seen in people diagnosed with SARS-CoV [106]. As observed in pathologic conditions such as COVID-19, MERS, and SARS, the infection not only affects the lungs but also impacts the regular cardiovascular and nephropathic rhythm, suggesting that the virus can indeed interfere with the vascular-endothelial cells and affect other parts of the body [107]. Even though ACE2 is present in the blood vessels, Monteil, (2020) showed that the COVID-19 virus could directly infect the blood vessels of engineered human calvarial osteoblasts (HCO) [86]. Moreover, Varga (2020) identified the presence of inflammatory cells and viral elements inside the endothelial cells of COVID-19-affected people, which eventually lead to cell death [108]. This information points out that SARS-CoV-2 infection has the ability to initiate endothelial inflammation at the multi-organ level, followed by pyroptosis, which may lead to host inflammatory response and cell injury [109]. Endothelial dysfunction can be associated with endothelial inflammation, vasoconstriction, and vascular lesions, which make diabetic individuals more prone to endothelitis of other organs; due to this, vasoconstriction may eventually lead to tissue edema, organ ischemia, and a pro-coagulant state [108]. Further studies are warranted to understand whether organ failure in COVID-19 is associated with direct viral infection or if any other factors are involved. Studies have shown that diabetic individuals having endothelial dysfunction may also contribute to pulmonary lesions and cytokine storms [110]. Glycemic oscillations have the ability to initiate the production of endothelial cytokine and adhesion molecules, which might cause unleashed extravasation of the leukocytes in the alveoli during the infection. It may eventually lead to impairment of respiratory function and lung damage [111,112,113]. Cytokine production from endothelial cells may contribute to pulmonary lesions in COVID-19 patients. Still, studies are needed to check whether this phenomenon is aggravated in individuals with and without diabetes.

### 4.5. Coagulopathy

Recent studies have shown that persons affected with COVID-19 and kept in ICU exhibited hyper-coagulation along with elevated levels of D-dimer and fibrin/fibrinogen-degraded products in multiple organs, which inversely affects the overall survival state [114,115,116]. Patients who suffered from severe COVID-19 also showed increased development of deep vein thrombosis and pulmonary embolism [117,118]. Active coagulation and host immune responses triggered by the infectious complications, in turn, activate multiple systemic coagulation [119]. The major reason for the occurrence of hypercoagulation is increased inflammatory response, which may be due to the COVID-19-associated “cytokine storm”. Diabetics are said to have more profound inflammatory responses, which may make them more prone to coagulation abnormalities. Further, T1D or T2D individuals show upregulation of markers of hyper-coagulation and fibrinolysis in plasma [120,121]. Under the hyperglycemic condition, coagulation and hyperinsulinemia were overstated and decreased fibrinolytic activity occurs during systemic inflammation [122]. The current study by Stegenga (2008) used the clamp technique in physically fit volunteers and elevated either insulin or glucose or both or none, followed by administration of a defined dose of LPS to cause a systemic inflammatory response [122]. After a certain period of time interval, hyperglycemic patients were shown to have more coagulation activity, and the degranulation of neutrophils was diminished. Hence, diabetics are likely to be more prone to thrombotic events occurring during an inflammatory response, but still, it remains unclear whether diabetic individuals experience increased coagulation during the period of COVID-19. Based on recent reports, SARS-CoV-2 does not appear to cause the major intrinsic pro-coagulant effects, whether SARS-CoV-2 or other consequences of cytokine storm precipitate the onset of systemic coagulation in COVID-19 patients will be critical in designing prevention and intervention therapies.

## 5. Role and Implications of Anti-Diabetic Drugs in the Clinical Management of SARS-CoV-2-Infected Diabetics

Diabetes mellitus is reported as one of the most commonly diagnosed co-morbidities in SARS-CoV-2-infected individuals [123]. Anti-diabetic drugs or glucose-lowering medications commonly used for the treatment of diabetes mellitus may also show indirect effects on COVID-19 pathogenesis, and these effects might have implications in the management of patients with diabetes mellitus and COVID-19. In the presence of mild SARS-CoV-2 infection in an outpatient setting, generally, glucose-lowering drugs for diabetes could be continued if the patient follows a prescribed dietary pattern and a more frequent blood glucose-monitoring regimen is implemented [124]. However, if the patients are admitted to the hospital with severe SARS-CoV-2 infection, it needs several modifications in their diabetes treatment strategies, including withdrawing the ongoing treatment and initiation of new therapies effectively. The results reported are mainly based on parameters such as the severity of SARS-CoV-2 infection, nutritional status, actual glycemic state and risk of hypoglycemia, renal function, drug interactions, etc.

### 5.1. Role of Insulin

Insulin has been widely used, and has been known for decades in patients hospitalized and suffering from diabetes. Continuous monitoring of glucose levels and decreased hypoglycemic rates positively correlated with insulin usage [125]. Insulin usage has also been one of the preferred treatment strategies for critically ill patients with DM amid the COVID-19 outbreak [22,90]. Furthermore, insulin showed reduced inflammatory marker levels in hospitalized patients with critical illness, playing an essential role as an anti-inflammatory agent [126]. Intravenous insulin administration showed beneficial effects on the suppression of coagulation and inflammation in hospitalized T2DM with COVID-19 [127]. Studies also found that, along with other severe infections, diabetic ketoacidosis (DKA) has also been reported in DM patients with COVID-19. Available evidence suggests that subcutaneous administration of insulin was an important treatment strategy for uncomplicated DKA in COVID-19 [128]. Chen et al. recently showed a positive correlation between diabetes and COVID-19, suggesting the need for monitoring COVID-19 patients with DM who are on regular insulin dosage [129]. A recent observational study reported that the requirement for insulin is significantly higher among COVID-19 patients [130], thus attributing to beta-cell dysfunction induced by SARS-CoV-2 infection.

Patients observed with poor oral intake and patients in mechanical ventilation require the intravenous administration of insulin infusion along with regular monitoring of glucose levels [131]. However, the adjustment and infusion rates will necessitate frequent visits to the patient and increased exposure to medical personnel. Hence, there is a significant need for exploration, and alternative insulin administration strategies. Previous studies have stated that the administration of subcutaneous short-acting insulin analogues as an alternate approach, as they have been used for several years, successfully, in converting mild to moderate diabetic ketoacidosis in physiological conditions [132,133]. Another alternate strategy is the usage of the single-dose administration of basal insulin. Although this approach has been successfully administered in critically ill patients in a study from Thailand to reduce regular contact with the patient [134], it still needs a specific approach. An insulin pump or continuous administration of subcutaneous insulin infusion (CSII) is another viable therapeutic option as some of these models have the advantage of altering insulin levels remotely by using Bluetooth [135]. Similarly, several fully automated closed-loop glucose level monitoring systems have been explored in critically ill patients [136], which can be experimented as alternative treatment strategies for treating patients with COVID-19 [137]. Moreover, insulin infusion, but not anti-hyperglycemic drugs, is the only option for COVID-19 patients with critically ill hyperglycemia.

### 5.2. Role of Metformin

The majority of T2DM patients use metformin alone or in combination with other drugs [138]. Several studies have reported that metformin increases the expression of ACE-2 in cells [139] and plays an important role in microvascular repair through adenosine monophosphate-activated protein kinase (AMPK) activation during acute lung injury [140]. A retrospective study stated that metformin decreased the mortality rates in type-2 diabetic patients, in particular, in women who were admitted to hospital with COVID-19 disease [141]. Mechanistically, metformin reduces pro-inflammatory cytokines such as TNFα and IL-6 while increasing anti-inflammatory cytokine IL-10 [142]. In addition, metformin exhibits a modulatory effect on mast cells and ACE2 expression, and improves endothelial function. These are the key mechanisms proposed to use metformin’s ability in reducing the severity of SARS-CoV-2 infection [141]. The use of metformin still continues in some exceptional cases such as in asymptomatic COVID-19 patients or patients with mild symptoms. However, continuous usage of metformin must be stopped in patients hospitalized with severe COVID-19 infection, as metformin is known to cause lactic acidosis [143].

### 5.3. Role of DPP4 Inhibitors

DPP4, commonly termed a cluster of differentiation 26 (CD26), is reported as a multifunctional transmembrane glycoprotein that plays a crucial role in glucose homeostasis and several inflammatory responses [144]. DPP4 plays an important role in the immune system as a T lymphocyte-activated marker and regulator of many chemokines such as CCL5, CXCL12, CXCL2, CXL11, etc. [145,146]. Mainly DPP4 is expressed in several organs such as the lung, spleen, kidney, liver, and immune cells [147]. Earlier studies reported that DPP4 inhibitors are widely used for the treatment of T2DM as they have negligible risk factors, such as hypoglycemia, and are well tolerated in the systemic circulation. A previous study reported that DPP4 is a functional receptor for MERS-CoV [148]; hence, it may also participate in SARS-CoV-2 infection despite other functions, and not being its primary entry receptor. Several studies also reported that targeting DPP4 has been considered a pharmacologically important strategy in patients suffering from severe respiratory diseases related to coronaviruses and COVID-19 [149,150]. DPP4 was also reported in the modulation of inflammation, and activation of immune functions [151]. Based on recent studies, ACE2 is mainly recognized as a key receptor for DPP-4 and also SARS-CoV-2 [152]. DPP-4 is predominantly expressed in the spleen, liver, lung, kidney, and some of the immune cells. Alterations in DPP4 expression were observed in T2DM. Moreover, some reports have stated that DPP4 is shed into the circulation as a soluble DPP4 [153,154,155]. Previous studies have shown that DPP4 inhibitor sitagliptin could decrease levels of pro-inflammatory markers such as tumor necrosis factor-α (TNF-α) and interleukin-6 (IL-6) [156,157]. Studies have reported that DPP4 inhibitors may have a role in the prevention of coronavirus primary infection and exhibit some anti-inflammatory effects [158]. In a multicenter, retrospective study including 338 patients with T2D and COVID-19, the administration of sitagliptin showed a decrease in mortality and improved clinical outcomes in patients [159]. In another retrospective study conducted on 904 patients with DM and moderate to severe COVID-19 patients, the administration of DPP4 inhibitors did not show a significant effect on patients’ mortality rates and clinical outcomes [129]. A recent epidemiological study stated that the administration of DPP4 inhibitors might not show any risk effect of hospitalization in 403 COVID-19 patients along with T2DM [160]. A previously conducted case study in Italy revealed that the association between DPP4 and COVID-19 and further administration of DPP4 inhibitors showed reduced mortality rates in 11 patients out of 387 [161]. In summary, treatment of T2DM with DPP4 inhibitors is associated with worse outcomes in twenty-seven patients who were treated with DPP4 inhibitors than in forty-nine patients who were on other glucose-lowering drugs [162]. Therefore, in-depth studies are still warranted to identify the potential survival benefits associated with the usage of DPP4 inhibition in patients with or without diabetes mellitus and COVID-19.

### 5.4. Role of GLP-1 Receptor Agonists

Glucagon-like peptide-1 receptor agonists (GLP-1 RA) are known to show a reduction in significant cardiac-related effects in T2DM patients [163]. Recent reports have stated that the administration of GLP-1 RA shows anti-inflammatory effects in low-grade inflammatory conditions such as atherosclerosis and non-alcoholic fatty liver disease [164,165]. GLP-1-RA may also contribute to (a) weight reduction in obese patients, and (b) a state of chronic inflammation compromised with several immune responses [166]. Previous studies have reported that administration of GLP-1RA shows a reduction in cytokine production and pulmonary inflammation in in-vivo studies [167]. Several preclinical studies have suggested that GLP-1 agonists show anti-inflammatory effects and might show pulmonary inflammation in rat studies with induction of experimental lung injury [168] and respiratory syncytial virus (RSV) infection [169]. Recent studies have hypothesized that GLP-1 agonists are mainly beneficial for COVID-19 patients and have a state of hyper-inflammation along with worse outcomes in people suffering from atherosclerosis, obesity, and T2DM. Recently several short-term studies have stated that patients in the ICU have shown that the administration of GLP-1RA medication is safe and effective in reducing blood glucose levels and management [170]. One study demonstrated the IL-6-lowering effect of GLP1 infusions in patients suffering from Type 1 DM [171]. Based on recent clinical data, controlling cardiovascular and renal functions are essential but challenging in COVID-19 patients. Recent studies have stated that the administration of GLP-1RA analogues is known to reduce several cardiovascular complications in T2DM patients, and shows anti-inflammatory effects, which might be useful for managing diabetes in SARS-CoV-2 infected patients [172,173].

Individuals with CVD or kidney diseases have shown worse prognoses during SARS-CoV-2 infection when compared with healthy individuals [114]. Therefore, it is important to identify the therapeutic approach for cardiorenal-implicated people who are at high risk of SARS-CoV-2 infection. Several studies have already reported the therapeutic roles of GLP1 analogues for the treatment of CVD and kidney diseases [173,174]. This drug administration is also a better option for treating diabetic subjects [175]. Recent studies have reported that being overweight and obese causes severe low-grade inflammation and shows a compromised immune system in diabetic patients [176]. Recent studies have stated that patients with COVID-19 and obesity-associated complications have worse health outcomes compared with COVID-19 alone. Healthcare providers have difficulties in identifying the correct mask size and associated problems with mask ventilation usage [176]. Because of their weight-reducing property, GLP1 analogues might be useful for patients with obesity and T2DM [177]. GLP-1-based drugs are effective pharmacological agents for targeting COVID-19 with and without diabetes. However, initiating such targeted therapies mostly in acute or critical situations is not well recommended because of several side effects [178]. Further, administration of GLP-1 receptor agonists showed a decrease in water intake [179]. Thus, GLP-1 agonist treatments are mainly associated with the risk of dehydration, and cause nausea, vomiting, and aspiration pneumonia in patients [180]. Based on studies, the administration of GLP-1 agonists was restricted in patients with severe COVID-19 [181]. If GLP1 agonists have to be administered to COVID-19 patients, they should be under close monitoring, and fluid intake should be adequate, etc. [143]. Administration of GLP-1 receptor agonists may cause aggravated anorexia; however, it was discontinued in several ill patients with COVID-19 because of the risk of aspiration pneumonia [143].

### 5.5. Role of SGLT2 Inhibitors

Sodium-glucose cotransporter-2 (SGLT2) inhibitors (SGLT2is) are known for the treatment of T2DM and show the effect on kidney SGLT2, which results in the reduction in blood glucose levels (shown in Figure 3). Administration of SGLT2 inhibitors to T2DM patients showed a reduction in the infiltration of several inflammatory cells into arterial plaques [182]. These inhibitors also showed a decrease in mRNA expression of inflammatory cytokines and several chemokines such as TNF-a, IL-6, and monocyte chemoattractant protein 1 (MCP1), etc. [183]. Previous reports have stated that the use of SGLT2 inhibitors plays a vital role in critically ill patients but causes ketoacidosis [184]. The use of SGLT2 inhibitors primarily showed the effects on urinary glucose and sodium excretion, which results in osmotic diuresis and potential dehydration [184], and increased urinary uric acid excretion, which is also correlated with a risk factor of acute kidney injury through urate crystal-dependent and independent mechanisms [185]. The use of SGLT2 inhibitors may be limited to several patients under critical conditions and illness, and patients have to be monitored closely as with the fluid state. SGLT2 inhibitors exert their antiviral efficacy by increasing lactate concentration and decreasing the intracellular pH, which results in lowering the viral load [186]. In COVID-19 patients, SGLT2-inhibitor canagliflozin showed a reduction in interleukin-6 (IL-6) levels, which are crucial in triggering cytokine release syndrome (CRS) [183]. A recent double-blinded, placebo-controlled phase III trial (DARE-19) (involving a total of 1250 patients) administered with dapagliflozin (10 mg daily or a placebo for 30 days) in patients hospitalized with SARS-CoV-2 infection showed cardiometabolic risk factors. However, the administration of SGLT-2 inhibitor did not show any reduction in organ dysfunction or death, and results showed no improved effects on clinical recovery regardless of its safety profile [187]. SGLT2 inhibitors are not recommended for critically ill patients with severe SARS-CoV-2 infection. However, there is no reason to limit SGLT2 inhibitors in patients with a mild form of SARS-CoV-2 infection [143].

### 5.6. Role of Thiazolidinediones

Thiazolidinediones are PPARγ agonists (peroxisome proliferator-activated receptor- γ). These agonists bind to nuclear receptors and undergo conformational changes, which results in the regulation and transcription of several genes involved in glucose and lipid metabolism [188]. Several in vitro and in vivo studies reported that administration of thiazolidinediones showed improvement in insulin resistance, activated putative anti-inflammatory and antioxidant effects, and contributed to the anti-atherosclerotic properties [189,190]. A recent study stated that thiazolidinediones showed the potential to mediate several protective effects on the cardiovascular system with unknown molecular mechanisms. However, long-term administration of thiazolidinediones promotes weight gain and edema in some patients [191]. More importantly, thiazolidinediones usage is positively associated with the aggravation of heart failure in patients [192]. Hence, due to these serious adverse effects of thiazolidinedione in patients with COVID-19, their usage is discouraged in recent times. Additional clinical trials and molecular evidence are warranted to optimize the risk-benefit ratio of thiazolidinediones use in patients with COVID-19.

### 5.7. Role of Sulfonylureas

Sulfonylureas are the oldest oral anti-diabetic drugs known for promoting insulin release from pancreatic β cells [193]. They work by closing the ATP-sensitive potassium channel, which results in the depolarization of the plasma membrane, increasing the calcium influx, which leads to insulin exocytosis within the cells [194]. Hypoglycemia is one of the serious adverse effects of this class of drugs [195]. Hence, it is suggested that the use of second-generation sulfonylureas such as glipizide, glibenclamide, tolbutamide, gliclazide, chlorpropamide, etc., should be avoided in patients with both DM and COVID-19 to prevent hypoglycemia, especially if oral intake is poor [196]. In addition, the simultaneous use of sulfonylureas along with chloroquine or hydroxychloroquine may increase the risk of hypoglycemia [90].

### 5.8. Role of Pioglitazone

Another classical oral anti-diabetic drug, pioglitazone, shows anti-inflammatory effects and anti-fibrotic effects in patients [197]. Previous studies revealed that pioglitazone up-regulates ACE2 expression in rat tissues [198], raising concerns about the possible increased susceptibility to SARS-CoV-2 infection [199,200]. Studies suggest that pioglitazone administration in diabetic patients with moderate SARS-CoV-2 infection might be helpful in preventing the cytokine storm [18,21]. Pioglitazone showed a decrease in the secretion of various pro-inflammatory cytokines in monocytes and macrophages [201]. Furthermore, Pioglitazone showed the inhibitory potential of blunting the cytokine storm by blocking the caspase recruitment at domain-containing protein-9 (CARD9), inhibiting the caspase activation in macrophages [202]. Recent computer-simulation-based bioinformatic study analysis revealed that pioglitazone might target the 3-chymotrypsin-like protease (3CLpro) and primarily inhibit SARS-CoV-2 RNA synthesis and replication [203]. Long-term usage of pioglitazone administration was positively associated with weight gain and edema in some patients, and, more importantly, aggravation of heart failure was observed [192], which did not support the pioglitazone use in patients with DM and COVID-19. Still, more clinical trials, molecular evidence, and clinical studies are warranted to optimize the risk-benefit ratio of pioglitazone in patients with COVID-19.

## 6. Effect of Long COVID on Diabetes Risk

Long COVID is the post-acute sequel, i.e., beyond the first 30 days of the acute phase of COVID-19 infection [204]. Recent studies have shown that long-COVID effects involve pulmonary and extra-pulmonary organ manifestations and contribute to the development of various diseases such as diabetes [204]. Yan xie MPH et al. (2022) stated that the post-acute phase of COVID-19 compared with the contemporary control group showed an increased risk of developing various non-communicable diseases such as CVDs, diabetes, etc. [205]. Several studies have also reported diabetes and glycometabolic abnormalities during the acute phase of COVID-19 infection. Furthermore, studies have also suggested an increased risk of diabetes in individuals aged 65 years or more, as well as in patients with a high BMI and area deprivation index quartiles [205]. In summary, the incidence of diabetes is much higher in individuals who have survived the first 30 days of acute SARS-CoV-2 infection, but, currently, it is not fully known about various molecular mechanisms that are responsible for this increase in diabetes in long-COVID cases. Accumulating results from recent studies point to the following mechanisms that might be responsible for the increased incidence of diabetes in long-COVID patients: (1) destruction of exocrine and endocrine cells due to SARS-CoV-2 infection; (2) transdifferentiation of pancreatic beta cells through the activation of eIF2 signaling pathway; and (3) induced auto-immunity and low-grade inflammation. However, conclusive studies are yet to be conducted to experimentally test these possibilities.

## 7. Effect of Pharmacological Drugs Used for COVID-19 Treatment on Glucose Metabolism

Some of the drug candidates used for the treatment of COVID-19 can also show the effect on glucose levels and glucose metabolism through the modulatory effects on inflammation and the immune system (Figure 4). Based on complications, these drugs may require particular considerations in clinical care and close monitoring when used in patients suffering from both COVID-19 and DM.

Camostat mesylate is a serine protease inhibitor that interferes with viral entry by targeting the transmembrane protease serine 2 (TMPRSS2), which is crucial for the facilitation of viral entry into the host cell [206]. Recent reports have stated that the administration of Camostat mesylate showed a decrease in the incidence of new-onset diabetes mellitus in patients with chronic pancreatitis [207]. Administration of Camostat mesylate shows the glycemic effect, insulin resistance, and decreased lipid accumulation in animal studies [208,209]. Despite their potential adverse effects, antimalarial drugs chloroquine (CQ) and hydroxychloroquine (HCQ) have been used for treating the SARS-CoV-2 infection [210,211]. Administration of HCQ showed two crucial mechanisms: (1) restriction of viral spike protein cleavage at specific ACE-2 binding sites, and (2) anti-inflammatory and immunomodulatory properties within the cells [212]. Recent studies have stated that hydroxychloroquine exhibited glucose-lowering efficacy through partially increasing insulin sensitivity and improving pancreatic β- cell functions [213]. Because of clinical relevance, hydroxychloroquine has been prescribed as anti-diabetic drug therapy in some countries [214]. Observational monitoring of pre-existing anti-diabetic drugs may be needed to avoid hypoglycemia conditions in patients with diabetes mellitus who regularly take hydroxychloroquine to treat COVID-19 [215,216,217]. Previous studies have stated contradictory results of hydroxychloroquine in the treatment of patients with COVID-19 [218,219].

Protease inhibitors such as Lopinavir and Ritonavir have already been reported to increase the risk of hyperglycemia, [220,221] new onset of DM cases [222], pre-existing diabetes mellitus and occasional development of diabetic ketoacidosis, etc. [223]. Recent reports have stated that the administration of Lopinavir and Ritonavir in HIV patients showed reduced insulin sensitivity and β-cell function by up to 50% [220]. Several studies have reported that Ritonavir acts as an inhibitor of the CYP3A4/5 enzyme [224], and inducer of uridine 5′- diphosphate-glucuronosyltransferase [225], which results in increasing plasma concentrations of the DPP4 inhibitor Saxagliptin, and decreasing concentrations of the SGLT2 inhibitor Canagliflozin [226]. Therefore, frequent monitoring of blood glucose levels and dosing pattern adjustments are highly suggested for patients treated with these types of drug combinations. Earlier studies stated that RNA-dependent RNA polymerase inhibitor Remdesivir showed improved hyperglycemia and insulin resistance, and fatty liver in mice models fed with a high-fat diet [227]. By contrast, the increase in blood glucose was similar between the Remdesivir-treated and placebo-treated groups in two randomized control trials (RCTs) with multi-ethnic groups [228] and Chinese patients [229]. Additional evidence is warranted to elucidate the Remdesivir effect on glucose metabolism.

IL-6 receptor inhibitors: IL-6 receptor inhibitors exert therapeutic options in patients suffering from severe COVID-19 illness who have extensive lung lesions with high IL-6 levels [230]. These inhibitors have shown beneficial effects on glucose intolerance and insulin resistance in patients with rheumatoid arthritis [231]. A recent study showed that Anakinra, an IL-1β inhibitor showed improved respiratory function in patients with severe COVID-19 [232]. A study also reported improved glycemia and β-cell function in patients with T2DM [233]. Furthermore, Canakinumab, another IL-1β inhibitor, is also under clinical trial for the treatment of COVID-19 [234]. However, it was not effective in COVID-19 patients with type 1 diabetes mellitus [234]. Several recent in vivo studies reported that Janus kinase 1/2 (JAK1/2) inhibitors and Bruton’s tyrosine kinase inhibitors were found effective for COVID-19 treatment [235], showed improved glycemia [236,237] and insulitis in patients [236,238,239]. Adalimumab, one of the TNF inhibitors, is a promising therapeutic option for mitigating the several inflammatory cytokines in COVID-19 [240]. Previous studies have reported that TNF inhibitors improved hyperglycemia, insulin resistance, and β-cell function in patients with active rheumatoid arthritis (RA) [241]. Systemic corticosteroid administration was found to be crucial against insulin resistance and pancreatic β-cell dysfunction [242,243,244]. A meta-analysis of clinical trials stated that systemic corticosteroid treatment is positively associated with reduced short-term all-cause mortality in patients with COVID-19 [245]. Administration of hydrocortisone in different regimens showed a significant tendency to produce a better hospital course in COVID-19 patients [246]. However, another recent study failed to prove the better effects of low-dose hydrocortisone treatment of COVID-19 patients [247]. A less-than-optimal dose regimen may be a reason for these negative results. Still, several investigations are required to prove the therapeutic effects of corticosteroids for the treatment of COVID-19 in patients associated with altered glucose metabolism.

## 8. Conclusions

The percentage of diabetic individuals is very high among SARS-CoV-2 infected patients. Currently, about 15% of COVID-19 patients are reported to have diabetes. The risk of developing COVID-19 is very high among diabetics. Even though the pathophysiological and molecular mechanisms connecting these two diseases are not fully understood, existing evidence has demonstrated that diabetes plays an important role in COVID-19 infection and affects the clinical severity of infections. One possibility for enhanced clinical severity of COVID-19 in diabetics is the deregulated immune cells, leading to poor immune responses to SARS-CoV-2 infection, co-existence of chronic conditions such as obesity, hypertension, and cardiovascular disease, and altered expression of ACE-2 receptors and, finally, endothelial dysfunction. Adherence to treatment recommendations in response to SARS-CoV-2 infection, regular blood glucose monitoring, and appropriate COVID-19 behaviors are highly recommended for patients with diabetes. The treatment practices for treating SARS-CoV-2-infected diabetics are different from those of the general population; hence, special care is required for COVID-19 patients with diabetes. Care must be exerted while recommending most commonly prescribed anti-diabetic drugs such as biguanides, SGLT2 inhibitors, DPP-4 inhibitors, sulfonylureas, and insulin as recent reports have observed that these drugs might exacerbate clinical conditions in COVID-19 cases. However, several controversies pertaining to the usage of commonly prescribed anti-diabetic agents for treating diabetic individuals infected with SARS-CoV-2 still persist as some studies report potential beneficial health effects while other ones show detrimental outcomes. Therefore, further research is needed to suggest an appropriate treatment and dietary regimen for individuals with co-existing diabetes and COVID-19.

## Figures and Tables

**Figure 1 pathogens-12-00184-f001:**
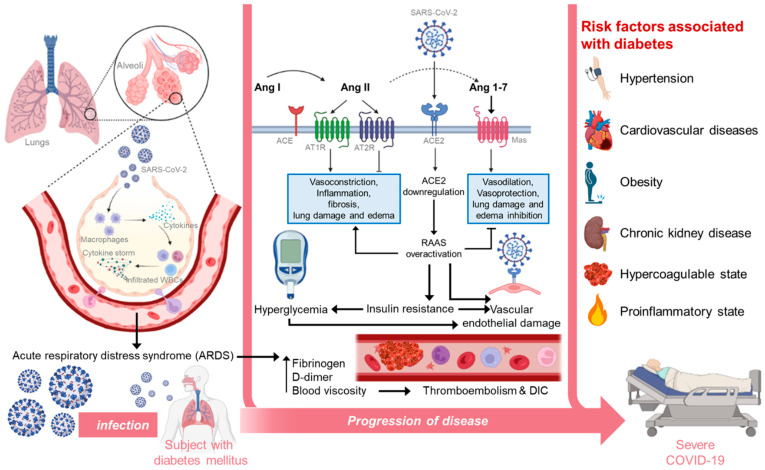
Clinical processes that are intensified after SARS-CoV-2 infection in individuals with diabetes mellitus. The figure was Created using BioRender.com (accessed on 26 December 2022).

**Figure 2 pathogens-12-00184-f002:**
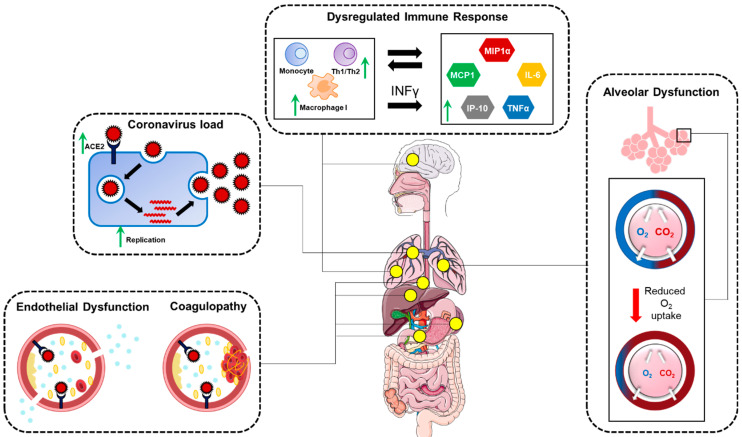
Potential mechanisms responsible for increased severity due to SARS-CoV-2 infection in individuals with diabetes. Human body organs. The artwork (Human body organs) was adopted from https://smart.servier.com/ (accessed on 23 June 2022).

**Figure 3 pathogens-12-00184-f003:**
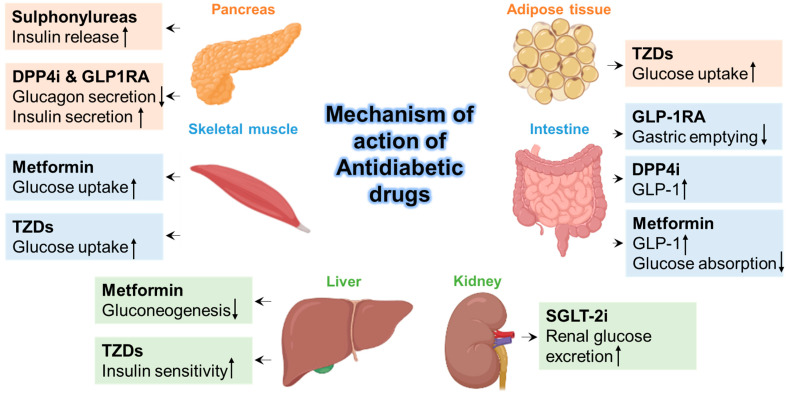
General mechanism of action of Antidiabetic drugs. The figure was Created using BioRender.com (accessed on 26 December 2022).

**Figure 4 pathogens-12-00184-f004:**
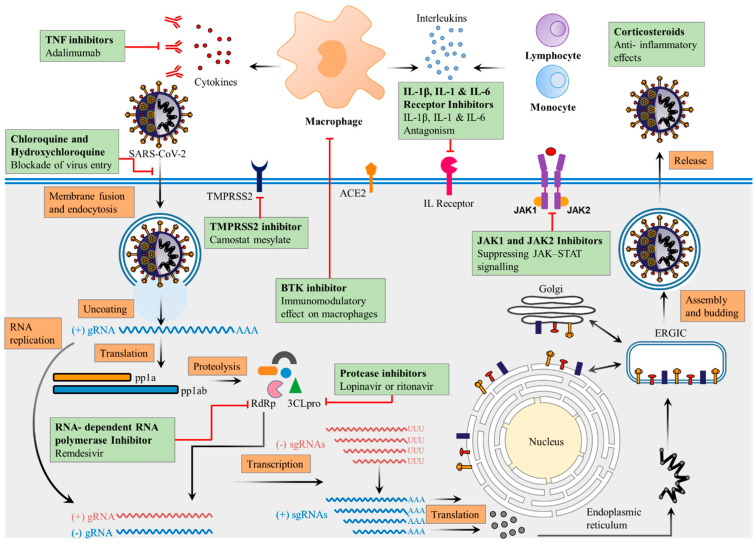
Drug candidates used for the treatment of COVID-19 that can affect glucose metabolism and their site of action.

## Data Availability

Not applicable.

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
