# Peer review of "COVID-19: Diabetes Perspective—Pathophysiology and Management"

_pathogens, 2023, doi:10.3390/pathogens12020184_

Round 1
Reviewer 1 Report
I appreciate the opportunity to review this interesting manuscript where the authors provide insights into the potential mechanisms linking COVID-19 and diabetes status. The authors present an extensive and inclusive analysis of the implications between both diseases. The manuscript is well structured and the writing is clear. The information presented addresses different perspectives on the research problem, which is why I believe it will be of great interest to Pathogens readers.
Author Response
Overall summary of the reviewer: I appreciate the opportunity to review this interesting manuscript where the authors provide insights into the potential mechanisms linking COVID-19 and diabetes status. The authors present an extensive and inclusive analysis of the implications between both diseases. The manuscript is well structured and the writing is clear. The information presented addresses different perspectives on the research problem, which is why I believe it will be of great interest to Pathogens readers.
Response: Authors would like to thank the reviewer for mentioning that the “manuscript is well structured, and the writing is clear” and for appreciating our work.
Reviewer 2 Report
The manuscript entitled "COVID-19: DIABETES PERSPECTIVE-PATHOPHYSIOLOGY AND MANAGEMENT". Title, abstract and overall rationale of work is well written and property explain. However, there are still some major concerns, which needs to be addressed before publication.
1) Abstract part is written very short and author must be elaborate this part. Moreover, diabetic is one word, no need to write abbreviation and remove it.
2) In the introduction section: Author wrote big paragraph without references and that is the important for giving the credit who describe these all first time. Secondly, author wrote in line number 67-68 several studies unfortunately author did not cite any articles (Several studies have shown that diabetics infected with Influenza A (H1N1), MERS-CoV, and SARS-CoV exhibit severe disease symptoms and succumb to death compared to non-diabetic individuals. Similarly, infection with SARS-CoV-2 is also associated with severe complications in diabetics.).
This introduction part is not focus, author just wrote general information and they did not write or focus on main topic. Author must be re-write the introduction section. Furthermore, author must be clarify why this review is important and novel for new researcher because I saw there are several review and research article available in the PubMed.
3) Section 2: Line no 75 author again said several studies but there are no any references available.
4) Section 3: Line no 95 this sentences is repeat in the above section author must be incorporate in the above section or delete it. Furthermore, in this section author talked about the what is the resion to increase the COVID-19 disease in the diabetic patient and they explain details while it is important here to add one figure to show details mechanism of all these and show there correlation/role in COVID-19 and diabetic.
5) Section 3.2: Line no.132-134 author describing about COVID-19 causes insulin resistance in patients and author should be explain here mechanism of action and author must be give references here that is important.
6) In the section 5.1, 5.2, 5.3,….. and 5.8: Author describe very well the role of insulin, role of metformin, role DPP4 Inhibitors and so on. It is important here author must be add one figure to show role of all these in one frame.
7) Conclusion section must be elaborate and this section should present at least in one 250-300 words paragraph and author must write future prospective and significance of this study.
8) There are some of punctuation and typographical errors throughout in the manuscript. Kindly correct it
Author Response
Overall summary of the reviewer: The manuscript entitled "COVID-19: DIABETES PERSPECTIVE-PATHOPHYSIOLOGY AND MANAGEMENT". Title, abstract and overall rationale of work is well written and properly explained. However, there are still some major concerns, which needs to be addressed before publication.
Query #1: Abstract part is written very short and authors must elaborate this part. Moreover, diabetic is one word, no need to write abbreviation and remove it.
Response: Authors would like to thank the reviewer for the comment. As per the reviewer's suggestions, the abstract is elaborated on in the revised submission.
Query #2: In the introduction section: Authors wrote big paragraph without references. It is important in giving the credit who describe these all first time. Secondly, author wrote in line number 67-68 several studies unfortunately author did not cite any articles (Several studies have shown that diabetics infected with Influenza A (H1N1), MERS-CoV, and SARS-CoV exhibit severe disease symptoms and succumb to death compared to non-diabetic individuals. Similarly, infection with SARS-CoV-2 is also associated with severe complications in diabetics).
This introduction part is not focussed, author just wrote general information and they did not write or focus on main topic. Author must re-write the introduction section. Furthermore, author must clarify why this review is important and novel for new researcher because I saw there are several reviews and research articles available in the PubMed.
Response: Authors would like to thank the reviewer for these comments and suggestions. New references are added in the first paragraph as appropriate and is rewritten. The introduction is more focussed in the revised submission. Moreover, we have explained in detail why this review is important in the current scenario.
Query #3: Section 2: Line no 75 authors again said several studies but there are no references available. Response: Authors would like to thank the reviewer for the comment and as suggested, references have been included as appropriate in the revised manuscript.
Query #4: Section 3: Line no 95 this sentences is repeat in the above section author must incorporate in the above section or delete it. Furthermore, in this section author talked about the reasons for increase in the COVID-19 disease in the diabetic patients, and they explain details while it is important here to add one figure to show details mechanism of all these and show there correlation/role in COVID-19 and diabetes. Response: Authors would like to thank the reviewer for this constructive comment. As suggested, a new figure is included in the revised manuscript to demonstrate the detailed mechanisms of correlation between COVID-19 and diabetes.
Query #5: Section 3.2: Line no.132-134 author describing about COVID-19 causes insulin resistance in patients and author should explain here mechanism of action and author must give references here that is important. Response: As per the reviewer’s suggestion, the mechanism of action describing COVID-19 causing insulin resistance has been added to the resubmitted version of the manuscript.
Query #6: In the section 5.1, 5.2, 5.3,….. and 5.8: Author describe very well the role of insulin, role of metformin, role DPP4 Inhibitors and so on. It is important here author must add one figure to show role of all these in one frame. Response: Authors would like to thank the reviewer for this critique. A new figure explaining the role of insulin, metformin, DPP4 inhibitors etc, is included in the revised manuscript.
Query #7: Conclusion section must be elaborate and this section should present at least in one 250-300 words paragraph and author must write future prospective and significance of this study. Response: Authors would like to thank the reviewer for this comment. The conclusions section has been elaborated, and the future prospects and significance of the study have been explained in the revised submission.
Query #8: There are some of punctuation and typographical errors throughout in the manuscript. Kindly correct it. Response: The typographical and punctuation errors have been corrected in the resubmitted manuscript.
Reviewer 3 Report
Dallavalasa and colleagues summarize some of the aspects relative to the increased risk of severe COVID-19 in patients with diabetes, along with the effect of drugs and, more broadly, some indications about the clinical management of these patients with the concomitant presentation of COVID and diabetes. The paper is plain, albeit some sentences are not exactly easy to read. On the other hand, it lacks novelty since dozens of papers have been published on the topic. Finally, some key important aspects must be improved.
1- As an example of strange sentences, in the abstract it is stated "..which have been shown to drive hyperglycaemia positively". A simpler "which have to shown to promote hyperglycaemia" is more appropriate. Please check carefully the text and modify accordingly.
2- The role of glycemic control in determining infection rates in vaccinated patients (and the relative immune response) and the severity of the disease in case of infection are not properly mentioned and discussed. Please see doi: 10.1038/s41467-022-30068-2. AND doi: 10.1002/dmrr.3476. discussing the results in detail.
3- the effect of long covid on diabetes risk could be added DOI: 10.1016/S2213-8587(22)00044-4
4- Also the section relatively to the effect of the drugs used to treat COVID in diabetes is incomplete and confusing. You might take advantage of published papers making a synthesis of the main findings on the topics. Please see doi: 10.1016/j.jdiacomp.2021.107927. and doi: 10.1016/S2213-8587(21)00244-8. and https://doi.org/10.1038/s41574-020-00435-4
5- Finally, it must be emphasized that in case of co-presentation of acute stress hyperglycaemia and severe COVID-19 (ICU admission), hyperglcyemia must be managed though insulin infusion and not just with drugs (see refs above + doi: 10.1186/s12933-020-01089-2), albeit data on this aspect are not conclusive
Author Response
Overall summary of the reviewer: Dallavalasa and colleagues summarized some of the aspects relative to the increased risk of severe COVID-19 in patients with diabetes, along with the effect of drugs and, more broadly, some indications about the clinical management of these patients with the concomitant presentation of COVID and diabetes. The paper is plain, albeit some sentences are not exactly easy to read. On the other hand, it lacks novelty since dozens of papers have been published on the topic. Finally, some key important aspects must be improved.
Query #1: As an example of strange sentences, in the abstract it is stated "..which have been shown to drive hyperglycaemia positively". A simpler "which have shown to promote hyperglycaemia" is more appropriate. Please check carefully the text and modify accordingly.
Response: As suggested by the reviewer, the sentences are written in simple and easy-to-understand language in the revised manuscript.
Query #2: The role of glycemic control in determining infection rates in vaccinated patients (and the relative immune response) and the severity of the disease in case of infection are not properly mentioned and discussed.
Response: Authors would like to thank the reviewer for the comment. The role of glycemic control in determining infection rates has been added as a new subheading 3.4 in the revised submission.
Query #3: The effect of long covid on diabetes risk could be added DOI: 10.1016/S2213-8587(22)00044-4.
Response: Authors would like to thank the reviewer for this suggestion. The effect of long COVID on Diabetes risk has been added as a new section (6).
Query #4: Also the section relatively to the effect of the drugs used to treat COVID in diabetes is incomplete and confusing. You might take advantage of published papers making a synthesis of the main findings on the topics. Please see doi: 10.1016/j.jdiacomp.2021.107927. and doi: 10.1016/S2213-8587(21)00244-8. and https://doi.org/10.1038/s41574-020-00435-4.
Response: Authors would like to thank the reviewer for this suggestion. The effect of drugs used to treat COVID in diabetes is rewritten in the revised article.
Query #5: Finally, it must be emphasized that in case of co-presentation of acute stress hyperglycaemia and severe COVID-19 (ICU admission), hyperglcyemia must be managed though insulin infusion and not just with drugs (see refs above + doi: 10.1186/s12933-020-01089-2), albeit data on this aspect are not conclusive.
Response: Authors thank the reviewer for this comment. Text related to this point has been included in the revised submission.
Round 2
Reviewer 2 Report
The authors have addressed all the concerns raised in the previous version of the manuscript and the quality has much improved after incorporating required modifications. Therefore, the manuscript may be considered for publication in this Journal.
Reviewer 3 Report
The manuscript is improved after the round of revision. I recommend to accept it in its present form.